# Control Efficacy of the Bt Maize Event DBN3601T Expressing Cry1Ab and Vip3Aa Proteins against Beet Armyworm, *Spodoptera exigua* (Hübner), in China

**DOI:** 10.3390/plants13141933

**Published:** 2024-07-14

**Authors:** Cheng Song, Xianming Yang, Limei He, Wenhui Wang, Kongming Wu

**Affiliations:** 1State Key Laboratory of Ecological Pest Control for Fujian and Taiwan Crops, Institute of Applied Ecology, Fujian Agriculture and Forestry University, Fuzhou 350002, China; songcheng990624@163.com; 2State Key Laboratory for Biology of Plant Diseases and Insect Pests, Institute of Plant Protection, Chinese Academy of Agricultural Sciences, Beijing 100193, China; yangxianming@caas.cn (X.Y.); w975480209@163.com (W.W.); 3Institute of Urban Agriculture, Chinese Academy of Agricultural Sciences, Chengdu 610299, China; helimei91@163.com; 4Institute of Insect Sciences, College of Agriculture and Biotechnology, Zhejiang University, Hangzhou 310058, China

**Keywords:** *Spodoptera exigua*, DBN3601T maize, susceptibility, pest management

## Abstract

The beet armyworm, *Spodoptera exigua* (Hübner), is a major pest of maize, cotton, soybean, and many other crops globally. Despite the widespread deployment of Bt transgenic maize for pest control worldwide, the efficacy of Bt lepidopteran-resistant transgenic maize in managing *S. exigua* remains rarely studied. In this study, we quantified the expression level of pyramided Cry1Ab and Vip3Aa toxins in Bt maize (event DBN3601T) and evaluated their control efficiency against *S. exigua* under both laboratory and field conditions. The enzyme-linked immunosorbent assay (ELISA) results showed that the expression levels of Cry1Ab and Vip3Aa proteins in DBN3601T maize tissues followed a decreasing order as follows: V5-leaf > V8-leaf > VT-tassel > R2-kernel > R1-silk. Diet-overlay assay results showed that the LC_50_ values of Cry1Ab and Vip3Aa proteins against *S. exigua* larvae were 11.66 ng/cm^2^ and 27.74 ng/cm^2^, respectively, with corresponding GIC_50_ values at 1.59 ng/cm^2^ and 7.93 ng/cm^2^. Bioassay using various tissues of the DBN3601T maize indicated that after 7 days of infestation, mortality rates of neonates and third-instar larvae ranged from 86% to 100% and 58% to 100%, respectively. Mortality was highest on V5 and V8 leaves, followed by R2-kernel, VT-tassel, and R1-silk. Field trials demonstrated that DBN3601T maize exhibited significantly lower larval density, damage rate, and leaf damage score compared to non-Bt maize. Field cage trial showed that the control efficacy of DBN3601T maize at the vegetative stage could reach 98%. These findings provide a theoretical basis for utilizing Bt transgenic maize to enhance the sustainable management of *S. exigua* in Asia.

## 1. Introduction

The beet armyworm, *Spodoptera exigua* (Hübner) (Lepidoptera: Noctuidae), native to southern Asia, is a polyphagous and migratory agricultural pest, active year-round in tropical regions [1,2,3]. This pest is known for causing severe infestations or outbreaks in China, especially south of the Yellow River Basin [3,4,5]. Its prevalence increases from north to south, manifesting five to six generations annually in the Yangtze River Basin, up to seven in warmer years, and reaching ten to eleven generations in Guangdong province [5,6]. At present, *S. exigua* has spread to more than 20 provinces (municipalities and autonomous regions) in China [6]. The *S. exigua* larvae have been documented infesting over 170 plant species, including maize, soybean, and cotton, leading to substantial losses to both food crops and economic crops such as vegetables [3,4,6]. China’s current strategy for the control of pests in maize fields is mainly based on spraying chemical insecticides during the vegetative growth stage. However, the long-term application of chemical pesticides has led to pest resistance and caused environmental pollution and food safety problems [7,8]. Resistance monitoring data showed that *S. exigua* has exhibited significant resistance to 10 different types of chemical insecticides, such as organophosphates, pyrethroids, and diamides [9,10]. Therefore, there is an urgent need to develop new efficient, and environmentally friendly technologies.

Bt (*Bacillus thuringiensis*) transgenic maize, which expresses specific insecticidal proteins, provides a new technical means for controlling major agricultural pests. In 1996, transgenic maize expressing the Cry1Ab proteins was commercialized in the United States, and its field control efficiency against lepidopteran pests such as *Ostrinia nublilalis* (Hübner) and *Diatraea grandiosella* (Dyar) was reported to exceed 99% [11,12,13]. Since then, countries like the United States, Canada, Argentina, and Brazil have widely adopted Bt maize, targeting lepidopteran and coleopteran pests such as *O. nublilalis*, *Spodoptera frugiperda* (J. E. Smith) and *Diabrotica virgifera virgifera* (LeConte) [14,15,16]. The widespread application of Bt maize has played a significant role in reducing chemical pesticide usage, controlling target pests, conserving natural enemies, and enhancing crop yields [17,18]. By 2022, the global area under Bt maize cultivation reached 62.2 million hectares, representing 32.7% of the total area planted with Bt crops [19].

The development of insect-resistant maize has progressed from early Bt maize varieties expressing single Cry protein to varieties with multiple Cry proteins. Currently, advanced pyramided Bt maize varieties that incorporate Cry proteins, Vip proteins, and RNA interference (RNAi) technologies are utilized to enhance efficacy and manage resistance [20]. Examples include Bt-(Cry1Ab+Vip3Aa20) Bt11×MIR162 and Bt-(Cry1Ab+Vip3Aa20+mCry3A) Bt11×MIR162×MIR604. The integration of Cry and Vip proteins in these cultivars broadens pest control and plays a vital role in integrated pest management (IPM) [21,22].

To enhance the management of maize pests, the Chinese government has been issuing biosafety certificates for the production and application of transgenic maize expressing various Bt insecticidal proteins such as Bt-Cry1Ab DBN9936, Bt-(Cry1Ab+Cry2Aj) Ruifeng 125, and Bt-(Cry1Ab+Vip3Aa) DBN3601T, aiming to accelerate the large-scale adoption of Bt maize [20]. Although these Bt maize varieties are not yet commercialized, their effectiveness in controlling principal pests such as *S. frugiperda*, *Mythimna separata* (Walker), *Helicoverpa armigera* (Hübner), and *Ostrinia furnacalis* (Guenée) has been extensively evaluated [23,24,25,26,27,28,29]. For example, DBN9936 maize exhibits high lethality to *S. frugiperda* and *M. separate* in leaf tissue, although its efficacy in silk and kernel tissue is moderate [25,26]. DBN9936 and Ruifeng 125 maize show high resistance to *O. furnacalis*, achieving insecticidal efficacy of over 98% [29]. Additionally, DBN3601T maize expressing pyramided Cry1Ab and Vip3Aa has demonstrated a control efficiency exceeding 95% against *S. frugiperda*, *M. separate*, and *Paralipsa gularis* (Zeller), indicating significant industrial potential in China [24,25,27]. However, research on the control efficacy of *S. exigua*, the key pest during the maize vegetative growth stage, remains notably insufficient.

With the promotion of commercial planting of Bt maize, in the absence of reliable stewardship, the effectiveness of Bt maize can be compromised by a rapid surge in Bt-resistant populations of the target pest [30]. For example, the inability of Bt-Cry3Bb1 maize to deliver high doses of Bt toxin, coupled with farmers’ limited compliance with structured refuge planting, has facilitated the rapid development of resistance to the *D. v. virgifera* [31]. In Puerto Rico, due to the annual planting of Bt maize and the lack of refuge, the *S. frugiperda* population rapidly developed resistance to Bt-Cry1F maize (event TC1507), leading to the early withdrawal of this product from the local market [32]. In China, pests remain susceptible to the main Bt insecticidal proteins expressed by Bt maize [24,25,33,34,35]. Bt maize can be used for pest control in China, but pest resistance management must be given adequate attention in regions where Bt maize is cultivated.

Hence, we assessed the susceptibility of a Chinese *S. exigua* population to the Cry1Ab and Vip3Aa proteins, evaluated the control efficacy of the combined Cry1Ab and Vip3Aa proteins in DBN3601T maize across various growth stages, and analyzed the control efficiency of DBN3601T maize on *S. exigua* in both laboratory and field settings. Our findings provide a data basis for integrated pest management and the development of target pest resistance strategies.

## 2. Results

### 2.1. Insecticidal Protein Expression in DBN3601T Maize Tissues

ELISA results confirmed that non-Bt maize tissue lacked the Cry1Ab or Vip3Aa insecticidal proteins. In DBN3601T maize tissues at various growth stages, the concentrations of Cry1Ab, Vip3Aa, and total Bt protein ranged as follows: 1.71~78.93 μg/g, 0.68~29.27 μg/g, and 2.39~108.20 μg/g, as presented in Table 1. Significant variations were observed in the expression levels of Cry1Ab, Vip3Aa, and total Bt protein across different tissues of DBN3601T maize (Cry1Ab: *F*_4,29_ = 7438.560, *p* < 0.001; Vip3Aa: *F*_4,29_ = 14,974.098, *p* < 0.001; total Bt protein: *F*_4,29_ = 11,714.772, *p* < 0.001). The Cry1Ab protein concentrations were consistently higher than that of Vip3Aa protein in the same tissue. The ranking of the Cry1Ab and total Bt protein expressions was as follows: V5-leaf > V8-leaf > VT-tassel > R1-silk > R2-kernel. For the Vip3Aa protein, the order was: V5-leaf, V8-leaf > VT-tassel > R2-kernel > R1-silk.

### 2.2. Susceptibility of Spodoptera exigua to Both Cry1Ab and Vip3Aa Proteins

Both Cry1Ab and Vip3Aa proteins exhibited high biological activity against *S. exigua*, with Vip3Aa demonstrating higher lethal and growth inhibition concentrations than Cry1Ab, as shown in Table 2. The LC_50_ values of Cry1Ab and Vip3Aa proteins against *S. exigua* larvae are 11.66 ng/cm^2^ and 27.74 ng/cm^2^, respectively, with LC_95_ values of 510.61 ng/cm^2^ and 756.59 ng/cm^2^. The GIC_50_ values of Cry1Ab protein and Vip3Aa protein on *S. exigua* larvae are 1.59 ng/cm^2^ and 7.93 ng/cm^2^, respectively, with GIC_95_ values of 144.14 ng/cm^2^ and 195.42 ng/cm^2^.

### 2.3. Lethal Effects of Various Tissues of DBN3601T Maize on Spodoptera exigua Larvae

The corrected mortality rates of *S. exigua* neonates residing on DBN3601T maize increased significantly over time (*F*_6,70_ = 854.060, *p* < 0.001) (Figure 1A). By day seven, the *S. exigua* neonates showed corrected mortality of 100.00% for V5-leaves and V8-leaves, 97% for VT-tassels, 86% for R1-silk, and 100% for R2-kernels of the DBN3601T maize. Significant differences in corrected mortality were observed across different tissues (*F*_4,70_ = 229.618, *p* < 0.001) (Figure 1A), ranking their lethality as follows: V5-leaf, V8-leaf > VT-tassel > R2-kernel > R1-silk.

The corrected mortality of third-instar *S. exigua* larvae residing on DBN3601T maize also increased significantly over time (*F*_6,70_ = 306.844, *p* < 0.001) (Figure 1B), peaking on the seventh day with 99% for V5-leaves, 100% for V8-leaves, 62% for VT-tassels, 58% for R1-silk, and 95% for R2-kernels. Significant mortality differences across the tissues were observed (*F*_4,70_ = 49.714, *p* < 0.001) (Figure 1B), with their lethality ranked as V8-leaf > V5-leaf, R2-kernel > VT-tassel, and R1-silk.

### 2.4. Field Control Efficacy of DBN3601T Maize on Spodoptera exigua Larvae by the Cage Trial

After infestation of maize plants with *S. exigua* at the V5 stage, the leaf damage score (Z = −5.568, *p* < 0.001), plant damage rates (Z = −5.574, *p* < 0.001), and the number of larvae per 100 plants (Z = −4.416, *p* < 0.001) in DBN3601T maize were significantly lower than those in non-Bt maize plants (Figure 2). Five days post-infestation, DBN3601T maize showed peak damage rates and scores at 18% and 0.18, respectively, compared to 89% and 2.44 for non-Bt maize (Figure 2A,B). On the first day after infestation, the number of larvae per 100 plants in DBN3601T maize and non-Bt maize was 718.67 and 622.00, respectively. On the third day after infestation, the number of larvae in DBN3601T maize rapidly decreased to 30.67. After 5 days of infestation, all larvae on DBN3601T maize plants died (Figure 2C). One day’s post infestation, the control efficacy of DBN3601T maize against *S. exigua* varied significantly across days (*F*_6,20_ = 6.836, *p* < 0.01) (Figure 2D). Control efficacy, measured by the reduction in plant damage rate, was 82% on the first day and reached 100% on the eleventh day.

## 3. Discussion

Bt insecticidal proteins exhibit efficient and specific insecticidal activity against a wide range of agricultural pests (such as lepidopteran, coleopteran, dipteran, etc.), but their toxicity levels can vary significantly [36]. Assessing pest susceptibility to Bt proteins and selecting appropriate Bt crops for pest management is crucial. Our study indicates that Cry1Ab and Vip3Aa proteins are particularly effective against *S. exigua*, with Cry1Ab showing high biological activity. Previous studies have shown that Cry1Da and Cry1Ca proteins demonstrated strong lethal and growth-inhibitory effects on *S. exigua*, while Cry1Ba and Cry2Ab proteins are non-toxic against this pest [37]. In the field, Bt-Cry1F cotton can effectively control the damage of *S. exigua*, while Bt-Cry1Ac cotton requires the application of insecticides to mitigate its damage, mainly because *S. exigua* is more susceptible to Cry1F protein [38,39]. Compared to lepidopteran pests such as *O. furnacalis* and *S. frugiperda*, Cry1Ab and Vip3Aa proteins exhibit stronger lethality against *S. exigua* in Henan and Anhui Provinces [34,35,40], further confirming that Bt maize expressing Cry1Ab or Vip3Aa proteins holds the potential for controlling *S. exigua*. Additionally, our results provide LC_50;95,_ and GIC_50;95_ values for Cry1Ab and Vip3Aa proteins against *S. exigua*, which may serve as benchmarks for early monitoring of field resistance to Bt maize after planting in China.

The pest-controlling effect of Bt maize mainly depends on the expression of its insecticidal proteins across its different tissues and growth stages [26]. Our observations indicated that the concentration of Bt protein expressed in DBN3601T maize follows the order leaves > tassels > kernels > silks. Specifically, the concentration of total Bt protein in leaf tissue was 13 times higher than in kernels and 44 times higher than in silk tissue. Similar results have been observed in Bt maize varieties MON810, DBN9936, MON88017, MIR162, and DKC64-24, where the concentrations of Cry1Ab, Cry3Bb1, Vip3Aa, Cry1F, and Cry2Ab2 proteins are significantly higher in leaf tissues compared to silks, tassels, stems, and kernels [26,41,42,43,44]. For instance, the concentration of Cry1Ab protein in the leaves of MON810 maize was found to be 4 to 20 times higher than in the roots and stems [43]. Similarly, the Cry1Ab protein content in the leaves of DBN9936 maize was the highest at (8.48 µg/g fresh weight), while in the kernels, the content was the lowest at 0.76 µg/g, representing an 11-fold difference [26]. In MIR162 maize, the expression levels of Vip3Aa protein in V6 leaves (228.0 µg/g dry weight) was six times that in R1 roots (40.2 µg/g) and twice that in kernels (115.0 µg/g) [44]. The high levels of Vip3Aa protein during the reproductive stage of MIR162 maize may be crucial for meeting the “high dose” requirement to effectively control *S. frugiperda* [26,28,44]. Furthermore, despite the Vip3Aa’s higher lethality against *M. separata*, Bt-Cry1Ab DBN9936 maize shows greater effectiveness than Bt-Vip3Aa DBN9501, due to significantly higher toxin levels (Cry1Ab, 76.54 µg/g dry weight) in its leaves compared to the toxin level (Vip3Aa, 5.08 µg/g) in DBN9501 maize leaves [25]. This underscores the critical role of high Bt protein expression in maize tissues for effective pest control. The concentration of Bt proteins in Bt maize is influenced by various factors, including environmental conditions (such as temperature and humidity), geographic location, and crop self-regulation [26]. For example, under cold/wet stimulation, the expression of Cry1Ab toxic protein in MON810 maize increased by four times [45]. Previous studies have measured the concentration of Vip3Aa protein in leaf tissues of DBN3601T maize grown indoors at 6.78 µg/g, which is 4.3 times lower than the concentration found in our field-grown leaves [23]. This discrepancy may be attributed to environmental factors such as temperature, humidity, and light exposure. Therefore, the insecticidal protein expression level of Bt maize should also be monitored in different locations and times under various field climates and cultivation environments.

This difference in Bt protein levels across tissues may lead to the exposure of target pests to medium and low dose levels, particularly during the reproductive stage, thereby increasing the probability of pest survival and affecting the field control effect of Bt maize. Our bioassay results revealed that the lethality of DBN3601T maize tissues to *S. exigua* neonates corresponds with the expression levels across different tissues, similar to findings for *S. frugiperda* and *M. separate* [25,27]. Our mesh cage trials showed that DBN3601T maize effectively controlled *S. exigua* during the vegetative growth stage, showing significantly lower leaf and plant damage compared to non-Bt maize, with larvae unable to survive on DBN3601T. In the southern United States, field-grown MON89034 and TC1507 maize significantly reduced leaf damage caused by *S. exigua* [46]. We also found that although the Bt protein content in the kernels was significantly lower than in the tassels, the kernels to third-instar larvae exhibited stronger lethality (95%) than the tassels (62%). Wang et al. also found that the lethality of kernel tissues of MON810 and Bt11 maize to *S. exigua* was significantly higher than that of the tassel [47]. This phenomenon may be due to the nutritional substances in maize kernels (such as high levels of lysine, methionine, and tryptophan) being unfavorable for the growth of *S. exigua* [48]. However, this hypothesis requires further investigation. Typically, *S. exigua* primarily inflicts damage on maize during the vegetative stages, but infestation is rare on the tassels or kernels during the reproductive stage in field conditions [49,50]. Hence, DBN3601T maize could serve as an effective tool for suppressing populations of *S. exigua* in China.

However, during the reproductive stage of maize, pests such as *S. frugiperda*, *O. furnacalis*, and *H. armigera* pose a greater threat. Zhao et al. indicated that DBN3601T maize plants at VT, R1, and R2 stages were more severely damaged and could screen out older larvae of *S. frugiperda* [27]. Our results also demonstrated that the tassels and silks of Bt maize exhibited lower control efficacy, especially to the third-instar larvae of *S. exigua*. This is likely due to insufficient Bt protein expression in the reproductive stage of maize. Nevertheless, the control efficacy of DBN3601T maize against *P. gularis* reached 100% during the R3-R6 stage, primarily due to the higher Cry1Ab protein levels, which exceeded the concentration needed to kill 95% of the insects. Conversely, the Vip3Aa protein expression in DBN3601T did not meet the necessary levels for effective control [24]. On the other hand, the substantial expression of Vip3Aa protein (99.9 µg/g dry weight) in the R6 stage of Bt11×MIR162 maize makes it more effective in controlling pests such as *S. frugiperda* compared to DBN3601T [12,44]. Zhang et al. demonstrated that DBN3601T maize tissues from the R1-R2 stage showed less efficacy against *M. separata*, yet they were more lethal than those of Bt-Cry1Ab DBN9936 and Bt-Vip3Aa DBN9501 maize [25]. The integration of Vip3Aa and Cry1Ab proteins in DBN3601T maize significantly improved its efficacy in pest control. However, the expression of Bt toxins in the reproductive stage is low, potentially compromising its long-term effectiveness. Variability of Cry1F expression among maize tissues can also impact the development and detection of resistance, as the dose expressed in vegetative stage plants may be higher than in reproductive stage plants. Pereira et al. observed that the survival of Cry1F-resistant *O. nubilalis* larvae was greater at the R1 stage of Bt-Cry1F maize compared to the vegetative growth stage [51,52]. Therefore, if DBN3601T maize were cultivated in China, significant attention should be given to controlling pests that exceed the economic threshold during the reproductive stage. This could be achieved through the use of biological control agents, environmentally friendly biopesticides, and other sustainable pest management practices.

Over the past 20 years, the cultivation experience with Bt crops has emphasized the importance of the ‘high-dose/refuge’ strategy in maintaining the effectiveness of Bt crops [53,54]. Refuge strategies typically include the incorporation of a certain percentage of non-Bt seeds into Bt seeds (seed-mixed refuge) or planting crops that do not express Bt toxins (structural refuge, natural refuge) in planting areas [55]. In the European Union, the European Food Safety Authority mandates that if a farm or a group of fields larger than 5 hectares cultivates MON810 maize, then 20% of the area must be planted with non-Bt maize as a refuge [15]. Additionally, the toxic protein concentration in the plants must be high enough to kill both susceptible and resistant heterozygous pests [15]. In the United States, using pyramided Bt maize to manage pests like *O. nublilalis* requires planting either a 5% structured refuge or a 5% seed-mixture refuge [20]. In the southern cotton-growing areas of the U.S., there are differing refuge requirements for Bt maize to manage resistance in *Helicoverpa zea* (Boddie), pyramidal Bt maize requires a 20% refuge area, while single-gene events like MON810 necessitate a 50% refuge [56,57]. In China’s small-scale Bt cotton agricultural systems, natural refuge crops, such as maize and soybean, are used to maintain susceptible populations of pests like *H. armigera*, which has a wide range of hosts and high mobility [53]. For pests with lower mobility, such as the *Pectinophora gossypiella* (Saunders), increasing the proportion of non-Bt cotton in seed-mixture refuges is necessary for resistance management [30].

Drawing from experiences in Bt crop cultivation and pest resistance management, China should adopt suitable resistance management strategies based on the type of Bt insecticidal genes, the biological characteristics of the pests, and the types of host plants. Considering that *S. exigua* often co-occurs with other lepidopteran pests such as *S. frugiperda*, *O. furnacalis*, and *M. separata*, it is recommended to conduct coordinated resistance monitoring and implement comprehensive resistance management strategies for these pests [58]. Firstly, determine the baseline susceptibility and resistance allele frequencies of target pests to Bt toxic proteins in different ecological zones. Establish practical resistance monitoring plans and assess the suitability of different Bt maize genotypes across various regions. Secondly, ensure that the Bt maize intended for widespread cultivation in China meets the high-dose standard for target pests. According to the 1998 Scientific Advisory Panel (SAP), a high dose is defined as 25 times the protein concentration necessary to kill susceptible larvae or achieve a 99.99% lethal dose [56]. If the Bt maize does not meet the high-dose standard, increasing the refuge area may be considered. Resistance to Cry1F maize by *O. nubilalis* in Canada illustrates that even high-dose Bt traits can be compromised by pest resistance and indicates the benefit of pyramiding Bt toxins, in conjunction with the use of non-Bt refuges, to bolster resistance management [59]. In DBN3601T maize has been shown to reach high dose levels of *S. frugiperda* and *O. furnacalis* in several development stages and at the R1 stage for *H. armigera* [28]. The results of this study indicate that the DBN3601T maize is highly effective against *S. exigua*, though whether it qualifies as a high-dose event is uncertain. It may be high-dose for the vegetative leaf feeding stage but not for the silk/kernel injury phase of maize. Thirdly, a refuge strategy should be implemented. Since *S. exigua* primarily damages maize at the seedling stage, pyramid Bt maize DBN3601T can be mixed with non-Bt maize, maintaining high toxin level at this stage. However, considering *S. exigua* often co-occurs with the primary target pest *S. frugiperda* in southern areas, the use of structural refuges is recommended in these regions. The high mobility of *S. frugiperda* between Bt plants and non-Bt plants in the seed-mixture scenario may increase the likelihood of resistance development [20]. In the northern region, where *S. frugiperda* is less common and *O. furnacalis*, which has low mobility but damages at the reproductive stage, predominates, structural refuges are still recommended due to low toxin expression at this stage for DBN3601T. Considering that small-scale farming is predominant in most regions of China, farmers are likely to prefer using Bt maize, which makes promoting structured refuges challenging. A refuge proportion of 10–20% is still recommended. Consequently, the Chinese government should disseminate knowledge about Bt maize, guide and supervise farmers’ planting practices, and support the implementation of the refuge strategy through subsidy policies. Finally, for migratory pests like *S. exigua* and *S. frugiperda*, plant polygenic pyramid Bt maize in both the source and landing areas of pest migration. Combine biological control methods such as light traps, pheromone traps, and sterile insect techniques to keep pest levels below economic damage thresholds.

## 4. Materials and Methods

### 4.1. Insects and Maize Materials

Following the method described by Guo et al. for identifying larval morphology [58], approximately 300 fourth- to fifth-instar *S. exigua* larvae were collected from a non-Bt maize field (22°38′12.29″ N, 101°52′19.01″ E) in Jiangcheng County, Yunnan Province, in April 2023. The collected larvae were fed with fresh maize leaves (V5 stage) indoors until pupation. After emergence, 10 females and 10 males were raised in plastic boxes (diameter: 12 cm; height: 9 cm) with a sterile gauze stopper and fed daily with a 10% honey–water solution (*v*/*v*). The tops of the boxes were covered with gauze for the adults to lay their eggs on, and the gauze with egg pieces was collected daily and placed in self-sealing bags for hatching. The artificial diet was in accordance with Xiao et al. [60]. Neonates (80 to 120 individuals) were transferred to plastic Petri dishes (90 mm diameter) using a brush for group rearing. After growth to third-instar larvae, they were transferred to a plastic cup (25 mL) containing an artificial diet using tweezers for single rearing until pupation. Both larvae and adults were reared at a temperature of 26 ± 1 °C, relative humidity of 60% ± 10%, and photoperiod of 16 h/8 h (L/D).

The seeds of Bt-(Cry1Ab+Vip3Aa) maize (event DBN3601T) and conventional hybrid Huaxingdan 88 with the same genetic background were supplied by Beijing DaBeiNong Biotechnology Co., Ltd., Beijing, China. For the tissue bioassay and field trials, maize was planted in June and October 2023 at Jiangcheng Experimental Station (22°31′15.57″ N, 101°29′44.54″ E), Jiangcheng County, in a plot area of 300 m^2^ (10 m × 30 m) and 25 m^2^ (5 m × 5 m). The spacing between the plots of different maize varieties was 1.5 m, with individual plants being spaced 35 cm apart in rows spaced 60 cm apart. Each experiment was repeated three times. Management practices included conventional tillage, a single urea application (approximately 150 kg/ha) at the V6 stage (the leaf pillow of the 6th leaf is visible), and no exposure to any insecticidal substances throughout the growing season.

### 4.2. Insecticidal Protein Expression in DBN3601T Maize Tissues

At the V5 (5th leaf collar stage), V8 (8th leaf collar stage), VT (tasseling stage), R1 (silking stage), and R2 (blister stage) growth stages of field maize plants [61], leaf tissues (youngest, 20 cm long), tassel tissues (closed, 5 cm long), silk tissues (unpollinated, 10 cm long), and kernel tissues (complete, with 30 capsules) were collected. During each sampling event, tissues were taken from three randomly selected plants in each plot, and this process was repeated three times. The collected plant tissue samples were placed in self-sealing bags and then pre-frozen in a freezer (−80 °C) for 24 h. The collected tissue samples were placed in a freeze-dryer (Shanghai Zhixin Experimental Instrument Technology Co., Ltd., Shanghai, China) until completely dry and then removed. The freeze-dried tissue samples were then ground into a fine powder using a pulverization mixer (Taikang Red Sun Electromechanical Co., Ltd., Taikang, China) and stored in a freezer (−80 °C) for preservation until needed. According to the measurement method of Wang et al. [23], and referring to the manufacturer’s instructions, the expression levels of Cry1Ab and Vip3Aa proteins in different tissues of DBN3601T maize were detected through sandwich ELISA using a Cry1Ab/Cry1Ac quantitative kit (Envirologix, Portland, OR, USA) and a Vip3A quantitative kit (YouLong Biotech, Shanghai, China). Data were presented as the amount of protein in dry weight of V5 leaf, V8 leaf, VT tassel, R1 silk, and R2 kernel.

### 4.3. Susceptibility of Spodoptera exigua to Cry1Ab and Vip3Aa Proteins

We used diet-overlay bioassays to measure the concentration response of *S. exigua* to Cry1Ab and Vip3Aa proteins. The artificial diet was aliquoted into a 24-well culture plate (hole diameter: 16 mm; height: 18 mm) (approximately 1 mL per well) and evenly spread to ensure a smooth surface without excess feed on the walls of the wells. Purified Cry1Ab or Vip3Aa proteins (ZhanNoSiTe Biotech, Beijing, China) were diluted to gradient concentrations of 0.00, 0.10, 0.24, 0.61, 1.54, 3.84, 9.60, 24.00, 60.00, and 150 ug/mL using distilled water. Each well of the culture plate was supplemented with 40 μL of the diluted toxin solution. For the control group, an equivalent volume of distilled water was used. The culture plate was gently shaken to evenly distribute the toxin solution over the feed surface and then allowed to air dry naturally. Each neonate (0–24 h) was gently transferred into individual wells using a soft-bristled brush, with 96 individuals tested for each concentration. The culture plate was sealed with Para-film and to prevent larvae from escaping and to maintain humidity. All plates were kept at a temperature of 26 ± 1 °C and a relative humidity of 60% ± 10% and with a photoperiod of 16 h/8 h (L/D). After feeding for 7 days, mortality rates were calculated, and the individual body weight of all larvae in the control and treatment groups were measured to calculate the weight inhibition rate. Larvae showing no response after a slight touch with the brush were considered dead.

### 4.4. Bioassay of DBN3601T Maize Tissue’s Resistance to Spodoptera exigua

Different plant tissues at various growth stages—V5 leaves, V8 leaves, VT tassel, R1 silks, and R2 kernels—were cut from the DBN3601T maize and non-Bt maize plants from a field trial and taken immediately to the laboratory for bioassays against *S. exigua* neonates and third-instar larvae. One type of maize tissue was cut into 1 cm segments (R2 kernels were complete) and placed in a 24-well culture plate. One neonate (0–24 h) or third-instar larvae (starved for 4–6 h) was transferred to each well with a fine brush, and the plates were sealed with Para-film to prevent the larvae from escaping and to keep them moisturized. Fresh maize plant tissues were replaced every day. Each treatment was replicated thrice for each maize variety, and 80 larvae were tested for each treatment, totaling 240 larvae. All plates were placed kept at a temperature of 26 ± 1 °C and a relative humidity of 60% ± 10%, with a photoperiod of 16 h/8 h (L/D). The dead and surviving larvae numbers were observed and recorded at 1 d, 2 d, 3 d, 4 d, 5 d, 6 d, and 7 d after inoculation. Mortality rates were calculated after 7 days. To ensure an accurate assessment of the insecticidal effects, mortality rates were corrected using Abbott’s formula to account for natural mortality observed in the control group [62]. Larvae showing no response after a slight touch with the brush were considered dead.

### 4.5. Field Control Efficacy of DBN3601T Maize on Spodoptera exigua Larvae by Cage Trial

After DBN3601T maize and non-Bt maize were sown, each plot (5 m × 5 m) was completely covered with 80 mesh gauze to isolate against interference from lepidopteran larvae such as *S. frugiperda* and *M. separate*. When maize plants reached the V5 stage, 50 maize plants were retained in each plot. Subsequently, 25 first-instar larvae were manually released into the whorl of each maize plant using a soft-bristled brush. After the artificial release of larvae, the plant damage rate, leaf damage score, and larval density were surveyed every other day. The damage-grading standard refers to the standard proposed by Williams et al., as shown in Table 3 [63]. After one day of larvae infection, the control effect of DBN3601T maize was calculated based on the plant damage rate. The test was repeated three times.

### 4.6. Data Analysis

Based on the experimental data, the mortality, corrected mortality, and control efficacy were calculated using Equations (1)–(3).
Mortality (%) = (number of dead test insects after treatment)/(number of test insects supplied before treatment) × 100(1)
Corrected mortality (%) = (treatment group mortality − control mortality)/(1 − control mortality) × 100(2)
Control efficacy (%) = (plant damage rate of non-Bt maize − plant damage rate of Bt maize)/(plant damage rate of non-Bt maize) × 100(3)

The susceptibility of *S. exigua* to Cry1Ab and Vip3Aa proteins was evaluated using probit regression to generate LC_50_, LC_95_, GIC_50,_ and GIC_95_ values with 95% fiducial limits. Differences in the expression levels of Bt toxic proteins in different tissues of DBN3601T maize and the control effect of DBN3601T maize were analyzed using one-way ANOVA, and multiple comparisons were made by using Tukey’s HSD if the differences were significant. The corrected mortality of DBN3601T maize on the *S. exigua* larvae was analyzed using two-way ANOVA, with multiple comparisons being made using Tukey’s HSD if the differences were significant. Mann–Whitney U test was used to analyze significant differences in leaf damage score, plant damage rate, and larval density between DBN3601T maize and non-Bt maize with the same infection days. Significance analysis of all data was performed using SPSS 26.0 (IBM, Armonk, NY, USA).

## 5. Conclusions

This study demonstrated that Bt-(Cry1Ab+Vip3Aa) maize DBN3601T can efficiently control the lepidopteran pest *S. exigua*, providing a new measure for the sustainable management of this pest in Asia. Planting of DBN3601T, even on a subset of cropping areas, may facilitate area-wide management of *S. exigua* and other polyphagous lepidopteran pests, as has been observed in cotton systems [64,65]. This can be achieved by integrating existing IPM strategies, such as pest monitoring and early warning, agriculture practices (e.g., reasonable rotation and intercropping), physical control (e.g., light traps), and biological control (e.g., parasitoid releases, sex pheromone traps, applications of baculoviruses and conservation biological control schemes [6,66]. Consequently, pest populations can be suppressed over a wide geographic area, leading to a sustainable intensification in maize crop growth.

## Figures and Tables

**Figure 1 plants-13-01933-f001:**
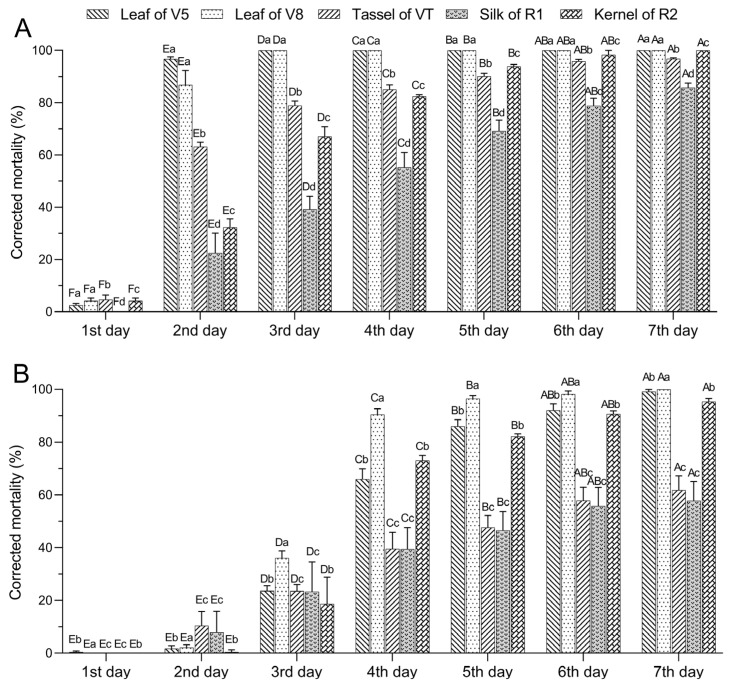
Corrected mortality of *Spodoptera exigua* neonates (**A**) and third-instar larvae (**B**) that fed on different tissues of DBN3601T maize for different durations. Error bars indicate the standard error of the mean. Different lowercase letters indicate significant differences in corrected mortality between different tissues, and different capital letters indicate significant differences in corrected mortality at different survey times (two-way ANOVA, Tukey’s HSD; *p* < 0.05).

**Figure 2 plants-13-01933-f002:**
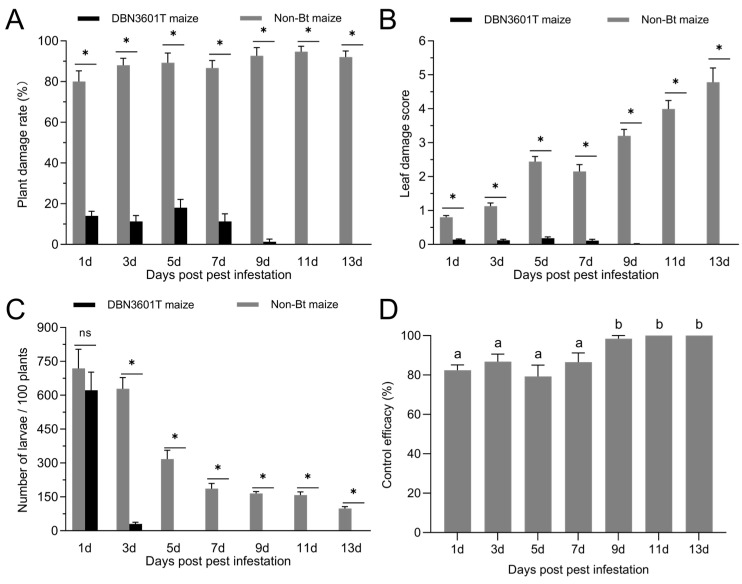
The control effects of DBN3601T maize against *Spodoptera exigua* in the field trial. (**A**) The plant damaged rate of Bt and non-Bt maize; (**B**) the average damage score of maize leaves; (**C**) the number of larvae per 100 maize plants; (**D**) the control effect of DBN3601T maize on *S. exigua* larvae. Error bars indicate the standard errors. Asterisks (*) denote significant differences in the rate of plant damage, leaf damage score, and the number of insects per 100 plants between the two maize varieties on the same days of infestation: * *p* < 0.05; ns indicates no significant difference (*p* > 0.05) (Mann–Whitney U test). Different lowercase letters indicate significant differences in the control efficacy of DBN3601T maize across various infestation durations (one-way ANOVA, Tukey’s HSD; *p* < 0.05).

**Table 1 plants-13-01933-t001:** Insecticidal protein expression in dry weight of DBN3601T maize tissues.

Growth Stage	Tissue	Cry1Ab (µg/g)	Vip3Aa (µg/g)	Total Bt Proteins (µg/g)
V5	Leaf	78.93 ± 0.89 a	29.27 ± 0.22 a	108.20 ± 0.98 a
V8	Leaf	76.33 ± 0.11 b	28.88 ± 0.09 a	105.21 ± 0.15 b
VT	Tassel	11.47 ± 0.19 c	4.83 ± 0.09 b	16.30 ± 0.22 c
R1	Silk	1.71 ± 0.19 e	0.68 ± 0.06 d	2.39 ± 0.17 e
R2	Kernel	4.43 ± 0.41 d	3.45 ± 0.04 c	7.88 ± 0.41 d

Values in this table are means ± standard errors. Different lowercase letters for data in the same column indicate significant differences in the Bt protein concentration between different tissues (one-way ANOVA, Tukey’s HSD; *p* < 0.05).

**Table 2 plants-13-01933-t002:** Lethal and growth inhibitory concentrations of Cry1Ab and Vip3Aa proteins to *Spodoptera exigua* larvae.

Bt Protein	N	Lethal Concentration				Growth Inhibitory Concentration				*df*
		LC_50_ (95%FL) ng/cm^2^	LC_95_ (95%FL) ng/cm^2^	Slope ± SE	*χ* ^2^	GIC_50_ (95%FL) ng/cm^2^	GIC_95_ (95%FL) ng/cm^2^	Slope ± SE	*χ* ^2^	
Cry1Ab	960	11.66 (8.67~15.18)	510.61 (332.41~879.81)	1.00 ± 0.07	21.57	1.59 (1.20~2.02)	144.14 (112.89~191.26)	0.84 ± 0.04	23.30	34
Vip3Aa	960	27.74 (20.86~35.83)	756.59 (500.61~1286.09)	1.15 ± 0.07	18.92	7.93 (7.01~8.90)	195.42 (162.02~241.02)	1.18 ± 0.04	13.40	34

N: number of insects tested. LC_50_ and GIC_50_ refer to the concentration of Cry1Ab and Vip3Aa proteins required to cause 50% mortality and 50% growth inhibition in the observation period of 7 days. Similarly, LC_95_ and GIC_95_ are the concentrations of Cry1Ab and Vip3Aa proteins required for 95% mortality and 95% growth inhibition.

**Table 3 plants-13-01933-t003:** Evaluation criteria for maize leaf damage by *Spodoptera exigua*.

Grade	Definition of Damage
0	No visible damage to leaves
1	Only pin-hole damage to leaves
2	Pin-hole and shot-hole damage to leaves
3	Small elongated lesions (5–10 mm) on 1–3 leaves
4	Midsized, elongated lesions (10–30 mm) on 4–7 leaves
5	Large, elongated lesions (>30 mm) or small portions eaten on 3–5 leaves
6	Large, elongated lesions (>30 mm) and large portions eaten on 3–5 leaves
7	Large, elongated lesions (>30 mm) and large portions eaten on 50% of leaves
8	Large, elongated lesions (>30 mm) and large portions eaten on 70% of leaves
9	Leaves destroyed on 70% of leaves

## Data Availability

Data are contained within this article.

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
