# Peer review of "Control Efficacy of the Bt Maize Event DBN3601T Expressing Cry1Ab and Vip3Aa Proteins against Beet Armyworm, Spodoptera exigua (Hübner), in China"

_plants, 2024, doi:10.3390/plants13141933_

Round 1

Reviewer 1 Report

Comments and Suggestions for Authors

Nice manuscript, well written, clear, logical steps.

Could the authors address, at least in some sentences in the conclusion, how this new solution could fit the exisiting maize IPM systems in China?

I suggest some minor changes, clarifications as per attached document.

Comments on the Quality of English Language

Author Response

Comments 1: Nice manuscript, well written, clear, logical steps.

Could the authors address, at least in some sentences in the conclusion, how this new solution could fit the exisiting maize IPM systems in China?

Response: Thank you for your positive feedback and suggestions for improvement. As suggested, the revised conclusion is presented below: “This study demonstrated that Bt-(Cry1Ab+Vip3Aa) maize DBN3601T can efficiently control the lepidopteran pest S. exigua, providing a new measure for the sustainable management of this pest in Asia. Planting of DBN3601T, even on a subset of cropping areas, may facilitate area-wide management of S. exigua and other polyphagous lepidopteran pests, as has been observed in cotton systems [64,65]. This can be achieved by integrating existing IPM strategies, such as pest monitoring and early warning, agriculture practices (e.g., reasonable rotation and intercropping), physical control (e.g., light traps), and biological control (e.g., parasitoid releases, sex pheromone traps, applications of baculoviruses and conservation biological control schemes [6,66]. Consequently, pest populations can be suppressed over a wide geographic area, leading to sustainable intensification of maize crops.” (lines 443-453)

Comments 2: I suggest some minor changes, clarifications as per attached document.

Title: The resistance evaluation of maize….. is a bit unusual. We usually use “control efficacy” or similar for the impact of the GM maize event XXX expressing YYYY protein(s) to control VVVV target pest(s).

The resistance evaluation reminds the readers on resistance allele frequency or resistance level(s) in target population.

General remark valid for many cases in the text, incl. title, abstract, etc.

Response: Accepted. The title has been revised to: “Control Efficacy of the Bt Maize Event DBN3601T Expressing Cry1Ab and Vip3Aa Proteins Against Beet Armyworm, Spodoptera exigua (Hübner) in China”. (lines 2-4)

Comments 3: L: 19 The wording “Bt transgenic maize” is correct in general insect pest management. However, if we talk on “lepidopteran pest control incl. S. exigua” the right wording is “Bt- lepidopteran resistant transgenic maize” or if talking on Diabrotica control “Bt-coleopteran resistant transgenic maize”.

Response: As suggested, we have now used the term "Bt-lepidopteran resistant transgenic maize" to accurately reflect the purpose. (line 19)

Comments 4: L: 33-34 Suggest, add specifically the event, otherwise the conclusion refers to all Bt transgenic maize.

Response: As suggested, we now use “DBN3601T maize” to replace “Bt-(Cry1Ab+Vip3Aa) maize”.

Comments 5: L 78-79: Industrial application is strange wording, probably large scale adoption or like that

Response: Accepted. We now use “large scale adoption”. (line 78)

Comments 6: L 100: toxin proteins might be replaced with toxic proteins, or toxins or proteins.

Response: Accepted. We now use “toxic proteins” instead.

Comments 7: L 108: Heading of Table 1 or in the text there is no indication, that the protein levels in various maize tissues are for dry weight? (or I overlooked?)

Response: We updated the title of Table 1 to “Insecticidal protein expression in the dry weight of DBN3601T maize tissues.” (line 120). Additionally, we have added a supplementary explanation in the method, which reads: “Data were presented as the amount of protein in dry weight of V5 leaf, V8 leaf, VT tassel, R1 silk, and R2 kernel.” (lines 378-379)

Comments 8: L 194: as already mentioned for the title, etc., authors may consider using “control efficacy, control success” or like that.

Response: Revised.

Comments 9: L 313: typo, breaking the sentence

Response: The sentence has been revised to: “Neonates (80 to 120 individuals) were transferred to plastic Petri dishes (90 mm diameter) using a brush for group rearing.” (lines 346-348)

Comments 10: L 321: typo (“m”)

Response: Revised.

Reviewer 2 Report

Comments and Suggestions for Authors

The authors are reporting on results of evaluating Bt proteins and Bt maize against the beet armyworm in China. This is a relatively thorough and straight forward study.  English is fine.  Several comments and suggestions:

Lines 112-145:  The authors provide data for neonates on diet, but neonates and third instars on plant tissue.  Did the authors conduct diet bioassays for third instars, and if not, why?

Lines 194-218:  This paragraph explains in great detail differences in Cry1Ab expression and possible results in toxicity.  But the authors do not seem to do the same for Vip3A which very likely is more toxic in planta than Cry1Ab.  Please add similar discussions for Vip3A that may include differences in toxicity on diet vs. in planta.

Also, although the authors discuss about expression of event DBN3601T in various tissues in regards to dose, efficacy and IRM concerns, there is no discussion (or data) the provides the contribution of each protein to overall dose and efficacy in this event which could have a considerable impact on overall durability.  Please discuss.

Line 283:  Please redefine "high dose", possibly using the definition cited by the Environmental Protection Agency (EPA)

Table3.  Really not needed.  Please discuss in text, and if possible, cite

Line 368:  The authors state that tissue was replaced every day.  Were these tissues that were being replaced from the same planting (therefore they were older), or were there subsequent plantings so that the tissue being replaced was the same leaf age?  If tissue being replaced from the same planting, does that impact expression values and interpretation? 

Author Response

Comments 1: Lines 112-145: The authors provide data for neonates on diet, but neonates and third instars on plant tissue. Did the authors conduct diet bioassays for third instars, and if not, why?

Response: Diet bioassays were not conducted on third-instar larvae, primarily because these tests are designed to assess larval susceptibility to toxic proteins by incorporating these proteins into the diet, typically focusing on neonate larvae. Furthermore, maize tissue assays were employed to evaluate the control efficacy of DBN3601T maize against both first- and third-instar larvae. However, conducting diet bioassays on third-instar larvae remains a valuable suggestion for future research to comprehensively assess the effectiveness of toxic proteins across different larval stages.

Comments 2: Lines 194-218: This paragraph explains in great detail differences in Cry1Ab expression and possible results in toxicity. But the authors do not seem to do the same for Vip3A which very likely is more toxic in planta than Cry1Ab. Please add similar discussions for Vip3A that may include differences in toxicity on diet vs. in planta. Also, although the authors discuss about expression of event DBN3601T in various tissues in regards to dose, efficacy and IRM concerns, there is no discussion (or data) the provides the contribution of each protein to overall dose and efficacy in this event which could have a considerable impact on overall durability. Please discuss.

Response: We greatly appreciate your valuable suggestion. In light of the potential for Vip3A to exhibit greater toxicity in plants than Cry1Ab, we have expanded our discussion to include similar analyses. The following has been incorporated into the manuscript: “In MIR162 maize, the expression levels of Vip3Aa protein in V6 leaves (228.0 µg/g dry weight) was 6 times that in R1 roots (40.2 µg/g) and twice that in kernels (115.0 µg/g) [44]. The high levels of Vip3Aa protein during the reproductive stage of MIR162 maize may be crucial for meeting the “high dose” requirement to effectively control S. frugiperda [26,28,44]. Furthermore, despite the Vip3Aa’s higher lethality against M. separata, Bt-Cry1Ab DBN9936 maize shows greater effectiveness than Bt-Vip3Aa DBN9501, due to significantly higher toxin levels (Cry1Ab, 76.54 µg/g dry weight) in its leaves compared to the toxin level (Vip3Aa, 5.08 µg/g) in DBN9501 maize leaves [25,26]. This underscores the critical role of high Bt protein expression in maize tissues for effective pest control.” (lines 212-221)

Furthermore, the contribution of the Cry1Ab and Vip3A proteins in DBN3601T to the overall dose and efficacy was discussed. In the DBN360T maize, the Cry1Ab and Vip3A proteins operate in conjunction to provide a comprehensive range of control effects and enhance the efficacy of pest control.

“Nevertheless, the control efficacy of DBN3601T maize against P. gularis reached 100% during the R3-R6 stage, primarily due to the higher Cry1Ab protein levels, which exceeded the concentration needed to kill 95% of the insects. Conversely, the Vip3Aa protein expression in DBN3601T did not meet the necessary levels for effective control [23]. On the other hand, the substantial expression of Vip3Aa protein (99.9 µg/g dry weight) in the R6 stage of Bt11×MIR162 maize makes it more effective in controlling pests such as S. frugiperda compared to DBN3601T [12,44]. Zhang et al. demonstrated that DBN3601T maize tissues from the R1-R2 stage showed less efficacy against M. separata, yet they were more lethal than those of Bt-Cry1Ab DBN9936 and Bt-Vip3Aa DBN9501 maize [26]. The integration of Vip3Aa and Cry1Ab proteins in DBN3601T maize significantly improved its efficacy in pest control. However, the expression of Bt toxins in the reproductive stage is low, potentially compromising its long-term effectiveness.” (lines 256-268)

Comments 3: Line 283: Please redefine "high dose", possibly using the definition cited by the Environmental Protection Agency (EPA).

Response: As suggested, we have revised the definition of “high dose” as follows: “a high dose is defined as 25 times the protein concentration necessary to kill susceptible larvae or achieve a 99.99% lethal dose” (lines 307-308)

Comments 4: Table 3. Really not needed. Please discuss in text, and if possible, cite.

Response: As suggested, we have removed the Table and incorporated the relevant information previously presented in Table 3 into the “Materials and Methods” section. (lines 363-366)

Comments 5: Line 368: The authors state that tissue was replaced every day. Were these tissues that were being replaced from the same planting (therefore they were older), or were there subsequent plantings so that the tissue being replaced was the same leaf age? If tissue being replaced from the same planting, does that impact expression values and interpretation?

Response: Accepted. We endeavored to ensure that the replaced tissue was of the same leaf age, potentially from different plantings, particularly for later days (e.g., 6-7 days). We believe that the toxin levels were not significantly different among various plantings, as indicated in Table 1, which includes three replications and demonstrates low standard error.

Reviewer 3 Report

Comments and Suggestions for Authors

Review of Song et al.: Evaluation of DBN36001T for control of S. exigua in maize

The study examines the activity of a pyramid of Cry1Ab + Vip3A maize against S. exigua in laboratory bioassays and a field experiment. The authors document protein expression in five tissues of maize plants and use diet overlay assays to calculate LC50 and GIC50 for each protein in a Chinese population of S. exigua. In a field experiment, the authors also document the mortality of S. exigua in plantings of maize expression Cy1Ab + Vip3A in comparison to a near isogenic non-Bt maize.  

The laboratory and field studies are well designed with adequate replication and use appropriate experimental procedures for the objective of each experiment. The results are analyzed and interpreted correctly.  This paper provides a useful and important contribution to our knowledge of how a Bt transgene preforms against S. exigua.  Indeed, no revision based on the research procedures and their presentation is needed. The manuscript is well written with only a small number of editorial corrections needed.  All the tables and figures are useful and needed.  The authors also cite relevant literature as needed with an appropriate number of references.

But one point is the use of the term ‘Resistance’ in the title. It seems the work described is not about resistance of S. exigua to Bt proteins but is about the efficacy or effectiveness of the Bt event to S. exigua in maize.  

A few minor editorial corrections and other comments are:

Line 40, ‘pest,’ should be ‘pest is’.

Line 89: change ‘the’ to ‘a’ Chinese…

Line 99: Suggest changing ‘were determined with the results detailed’ to ‘are presented’ in Table 1.

Line 182 add ‘a’ between from and Chinese.

Line 182 and 184; ‘range’ should be ‘ranged’.

Line 262-264.  In the southern cotton-growing areas of the U.S., there is a 20% refuge requirement for pyramided Bt products and a 50% refuge requirement for single gene events like MON810.

Lines 292-294.  The results presented in this study show that the DBN3601T event in maize is highly effective against S. exigua.  Whether it is a ‘high-dose’ event is uncertain.  It may be high-dose for the vegetative leaf feeding stage but not for the ear/kernel injury phase of maize.

Line 296-298. Considering the tendency for S. exigua to develop resistance to conventional insecticides as cited on lines 51 to 53, some sort of Bt refuge management seems needed to prolong the efficacy of DBN3601T in maize in China. What the best level of refuge is and whether it should be a structured or seed-mixture is difficult to determine. Seed mixture ensures refuge compliance, but larval movement may expose larvae to sub-lethal doses of the toxins. The type of refuge and levels non-Bt maize in a refuge must consider all of a crop’s primary target pests.   

Comments on the Quality of English Language

The manuscript is very well written.  I listed several minor corrections in my comments. 

Author Response

Comments 1: But one point is the use of the term ‘Resistance’ in the title. It seems the work described is not about resistance of S. exigua to Bt proteins but is about the efficacy or effectiveness of the Bt event to S. exigua in maize.

Response: As suggested, we have now revised the title to: “Control Efficacy of the Bt Maize Event DBN3601T Expressing Cry1Ab and Vip3Aa Proteins Against Beet Armyworm, Spodoptera exigua (Hübner) in China”. (lines 2-4)

Comments 2: Line 40, ‘pest,’ should be ‘pest is’.

Response: Accepted.

Comments 3: Line 89: change ‘the’ to ‘a’ Chinese.

Response: Accepted.

Comments 4: Line 99: Suggest changing ‘were determined with the results detailed’ to ‘are presented’ in Table 1.

Response: Revised. we used “as presented in Table 1” to replace it. (line 112)

Comments 5: Line 182 add ‘a’ between from and Chinese.

Response: Revised.

Comments 6: Line 182 and 184; ‘range’ should be ‘ranged’.

Response: Revised.

Comments 7: Line 262-264. In the southern cotton-growing areas of the U.S., there is a 20% refuge requirement for pyramided Bt products and a 50% refuge requirement for single gene events like MON810.

Response: We greatly appreciate your valuable suggestion. We have added the following information “In the southern cotton-growing areas of the U.S., there are differing refuge requirements for Bt maize to manage resistance in Helicoverpa zea (Boddie), pyramidal Bt maize requires a 20% refuge area, while single-gene events like MON810 necessitate a 50% refuge [56,57].” (lines 287-290)

Comments 8: Lines 292-294. The results presented in this study show that the DBN3601T event in maize is highly effective against S. exigua. Whether it is a ‘high-dose’ event is uncertain. It may be high-dose for the vegetative leaf feeding stage but not for the ear/kernel injury phase of maize.

Response: As suggested, this sentence has been revised to: “The results of this study indicate that the DBN3601T maize is highly effective against S. exigua, though whether it qualifies as a high-dose event is uncertain. It may be high-dose for the vegetative leaf feeding stage but not for the silk/kernel injury phase of maize.” (lines 314-317)

Comments 9: Line 296-298. Considering the tendency for S. exigua to develop resistance to conventional insecticides as cited on lines 51 to 53, some sort of Bt refuge management seems needed to prolong the efficacy of DBN3601T in maize in China. What the best level of refuge is and whether it should be a structured or seed-mixture is difficult to determine. Seed mixture ensures refuge compliance, but larval movement may expose larvae to sub-lethal doses of the toxins. The type of refuge and levels non-Bt maize in a refuge must consider all of a crop’s primary target pests.

Response: We greatly appreciate your suggestion. We have revised the text as follows:

“Thirdly, a refuge strategy should be implemented. Since S. exigua primarily damages maize at the seedling stage, pyramid Bt maize DBN3601T can be mixed with non-Bt maize, maintaining high toxin level at this stage. However, considering S. exigua often co-occurs with the primary target pest S. frugiperda in southern areas, the use of structural refuges is recommended in these regions. The high mobility of S. frugiperda between Bt plants and non-Bt plants in the seed-mixture scenario may increase the likelihood of resistance development [20]. In the northern region, where S. frugiperda is less common and Ostrinia furnacalis, which has low mobility but damages at the reproductive stage, predominates, structural refuges are still recommended due to low toxin expression at this stage for DBN3601T. Considering that small-scale farming is predominant in most regions of China, farmers are likely to prefer using Bt maize, which makes promoting structured refuges challenging. A refuge proportion of 10%-20% is still recommended. Consequently, the Chinese government should disseminate knowledge about Bt maize, guide and supervise farmers’ planting practices, and support the implementation of the refuge strategy through subsidy policies.”. (lines 317-331)

Reviewer 4 Report

Comments and Suggestions for Authors

Song et al. explore the resistance evaluation of Bt-(Cry1Ab+Vip3Aa) maize (Event 2 DBN3601T) to the beet armyworm, Spodoptera exigua (Hübner), in China. The study is significant and would be a valuable contribution to those working in the field of pest resistance and agricultural biotechnology. However, I have found numerous issues throughout the manuscript, including poorly organized sections, contradictory statements, and a lack of clarity in the methodology and results. Additionally, the English language used in the paper requires extensive revision by experts who have subject knowledge. Therefore, I recommend a major revision to improve its quality before final acceptance.

Comments on the Quality of English Language

Moderate editing of English language required.

Author Response

Comments 1: Abstract: The abstract is very well written, covering all aspects of the study in a comprehensive manner. However, I have a minor suggestion: instead of writing precise percentages like 27.74% or 100.00%, I recommend using rounded figures such as 28% or 100%, respectively.

Response: Accepted.

Comments 2: Introduction: The current section is poorly organized with information presented randomly and some sentences contradicting each other. The hypothesis and research gap are not properly explained. The section needs a systematic flow of information, and a transition paragraph is missing.

Response: We appreciate the reviewer's insightful comments regarding the organization and clarity of the Introduction section. Following your suggestions, we have extensively revised the Introduction to enhance its structure and coherence. Please refer to the revised Introduction section.

Comments 3: Line 40: This pest is known.

Response: Revised.

Comments 4: Please be consistent in using either “Beet armyworm” or “S. exigua.”

Response: Accepted. We now use “S. exigua” consistently.

Comments 5: Lines 53-55: Bt crops have been in use for ages. The authors have acknowledged their use in agriculture against pests. Therefore, instead of this statement, the authors should highlight the main purpose of conducting their study as mentioned in Line 19. I recommend reconsidering this statement.

Response: We greatly appreciate your suggestion. Considering the need to introduce new technical means for the control of S. exigua in China, we re-describe this sentence as: “Therefore, there is an urgent need to develop new efficient, and environmentally friendly technologies.” (lines 53-54)

Regarding the main research purpose of the thesis, we mention it at the end of the introduction: “However, research on the control efficacy of S. exigua, the key pest during the maize vegetative growth stage, remains notably insufficient.” (lines 88-89)

Comments 6: Lines 67-72: This is very concerning. On one hand, the authors recommend enhancing the adoption of Bt maize to reduce reliance on chemical pesticides. On the other hand, they state that ‘dozens of pests have developed substantial resistance to Bt maize.’ Please thoroughly revise the text to avoid contradiction.

Response: Accept. We have revised the text and structure: “With the promotion of commercial planting of Bt maize, in the absence of reliable stewardship, the effectiveness of Bt maize can be compromised by a rapid surge in Bt-resistant populations of the target pest [30]. For example, the inability of Bt-Cry3Bb1 maize to deliver high doses of Bt toxin, coupled with farmers’ limited compliance with structured refuge planting, has facilitated the rapid development of resistance to the D. v. virgifera [31]. In Puerto Rico, due to the annual planting of Bt maize and the lack of refuge, the S. frugiperda population rapidly developed resistance to Bt Cry1F maize (event TC1507), leading to the early withdrawal of this product from the local market [32]. In China, pests remain susceptible to the main Bt insecticidal proteins expressed by Bt maize [24,25,33–35]. Bt maize can be used for pest control in China, but pest resistance management must be given adequate attention in regions where Bt maize is cultivated.” (lines 90-100)

Comments 7: Lines 79-81: This sentence seems out of place and irrelevant.

Response: As suggested, we have removed this sentence.

Comments 8: Lines 82-88: This section is very confusing and repetitive. It is unclear what conclusion the authors want to make from this information.

Response: We greatly appreciate your valuable feedback. The revised section now reads: “Although these Bt maize varies are not yet commercialized, their effectiveness in controlling principal pests such as S. frugiperda, Mythimna separata (Walker), Helicoverpa armigera (Hübner), and Ostrinia furnacalis (Guenée) has been extensively evaluated [23–29]. For example, DBN9936 maize exhibits high lethality to S. frugiperda and M. separate in leaf tissue, although its efficacy in silk and kernel tissue is moderate [25,26]. DBN9936 and Ruifeng 125 maize show high resistance to O. furnacalis, achieving insecticidal efficacy of over 98% [29]. Additionally, DBN3601T maize expressing pyramided Cry1Ab and Vip3Aa has demonstrated a control efficiency exceeding 95% against S. frugiperda, M. separate, and Paralipsa gularis (Zeller), indicating significant industrial potential in China [24,25,27]. However, research on the control efficacy of S. exigua, the key pest during the maize vegetative growth stage, remains notably insufficient.” (lines 79-89)

Comments 9: Line 88: What is meant by “Bt maize event”?

Response: We believe the term "Bt maize event" refers to a specific Bt maize variety. In the manuscript, we have replaced it with “DBN3601T” for greater precision.

Comments 10: Transition paragraph is missing.

Response: Accepted. We have revised the structure and made it more fluent. The revised paragraph is as follows: “The development of insect-resistant maize has progressed from early Bt maize varieties expressing single Cry protein to varieties with multiple Cry proteins. Currently, advanced pyramided Bt maize varieties that incorporate Cry proteins, Vip proteins, and RNA interference (RNAi) technologies are utilized to enhance efficacy and manage resistance [20]. Examples include Bt-(Cry1Ab+Vip3Aa20) Bt11×MIR162 and Bt-(Cry1Ab+Vip3Aa20+mCry3A) Bt11×MIR162×MIR604. The integration of Cry and Vip proteins in these cultivars broadens pest control and plays a vital role in integrated pest management (IPM) [21,22].” (lines 67-74)

Comments 11: Materials and Methods: Details are missing and require extensive revision to be clear and understandable for readers.

Response: We have revised the section to include a detailed description of the methods of obtaining materials, such as the collection and feeding of insects, and the collection of Bt/non-Bt tissue samples. Additionally, we have supplemented and redescribed the experimental methods for DBN3601T maize tissue insecticidal protein expression and both the bioassay.

Comments 12: Lines 305-308: Please rephrase these sentences.

Response: As suggested, we have now amended this sentence to: “Following the method described by Guo et al. for identifying larval morphology [58], approximately 300 fourth- to fifth-instar S. exigua larvae were collected from a non-Bt maize field (22°38'12.29"N, 101°52'19.01"E) in Jiangcheng County, Yunnan Province, in April 2023.” (lines 338-341)

Comments 13: Line 309: Instead of “100 ml,” mention the dimensions of the base (width and length).

Response: Accepted. This has been revised to “plastic boxes (diameter: 12 cm; height: 9 cm)”. (lines 342-343)

Comments 14: Line 314: How many neonates were put in one Petri dish? “Petri dishes with a diameter of 90 mm” can be written as “Petri dishes (90 mm diameter).”

Response: As suggested, we have revised the sentence to “Neonates (80 to 120 individuals) were transferred to plastic Petri dishes (90 mm diameter) using a brush for group rearing.” (lines 346-348)

Comments 15: Line 319: non-Bt near-isogenic maize… Is near-isogenic the name of the variety?

Response: “near-isogenic” is not a name of the variety. This has been revised to “conventional hybrid Huaxingdan 88 with the same genetic background”. (lines 352-353)

Comments 16: Line 321: For the tissue bioassay and field trials. maize.....please remove the full stop.

Response: Revised.

Comments 17: Line 321: “11 June, and 1 October” - I recommend mentioning only the month and year.

Response: Revised.

Comments 18: Line 326: Please elaborate on the “V6 stage” with a single explanatory sentence.

Response: As suggested, we have added an interpretation of the “V6 stage” in the manuscript, i.e., the leaf pillow of the 6th leaf is visible.

Comments 19: Line 330: Explaining the abbreviations (V5, V8, VT, R1, R2) will help readers understand better.

Response: As suggested, we cited the literature to explain it. This is now changed to “At the V5 (5th leaf collar stage), V8 (8th leaf collar stage), VT (tasseling stage), R1 (silking stage), and R2 (blister stage) growth stages of field maize plants [61]”. (lines 363-364)

Comments 20: Line 332: It should be “freezer” instead of “refrigerator.”

Response: As suggested, we have changed “refrigerator,” to “freezer”. (line 369)

Comments 21: Line 334: “Fully ground using a tissue grinder” is confusing. Please rephrase this sentence.

Response: Revised. The sentence is now amended to “The freeze-dried tissue samples were then ground into a fine powder using a pulverization mixer (Taikang Red Sun Electromechanical Co., LTD. Taikang, China) and stored in a freezer (–80°C) for preservation until needed.” (lines 371-374)

Comments 22: Line 342: Improve the description of the “concentration-response bioassay using the diet-overlay method.”

Response: Revised. The sentence is now described as “We used diet-overlay bioassays to measure the concentration response of S. exigua to Cry1Ab and Vip3Aa proteins.” (lines 382-383)

Comments 23: Lines 350-352: Rewrite this sentence.

Response: Revised. The sentence is now described as “Each neonate (0–24 h) was gently transferred into individual wells using a soft-bristled brush, with 96 individuals tested for each concentration.” (lines 391-393)

Comments 24: Lines 374-375: Improve this part.

Response: Revised. The sentence is now described as “Larvae showing no response after a slight touch with the brush were considered dead.” (line 398)

Comments 25: Results: I recommend revising the presentation of results. Starting with a table reference does not seem ideal. Provide the results first and then refer to the relevant table. The section needs thorough revision for better representation of findings.

Response: As suggested, we have revised the presentation of the results for sections “2.1” and “2.2” for better clarity and flow. These sections now read:

“ELISA results confirmed that non-Bt maize tissue lacked the Cry1Ab or Vip3Aa insecticidal proteins. In DBN3601T maize tissues at various growth stages, the concentrations of Cry1Ab, Vip3Aa, and total Bt protein ranged as follows: 1.71~78.93 μg/g, 0.68~29.27 μg/g, and 2.39~108.20 μg/g, as presented in Table 1. Significant variations were observed in the expression levels of Cry1Ab, Vip3Aa, and total Bt protein across different tissues of DBN3601T maize (Cry1Ab: F4, 29 = 7438.560, p < 0.001; Vip3Aa: F4, 29 = 14974.098, p < 0.001; total Bt protein: F4, 29 = 11714.772, p < 0.001).” (lines 109-115)

“Both Cry1Ab and Vip3Aa proteins exhibited high biological activity against S. ex-igua, with Vip3Aa demonstrating higher lethal and growth-inhibition concentrations than Cry1Ab, as shown in Table 2.” (lines 125-127)

Comments 26: Line 99: Delete “with the results detailed in.”

Response: Revised. We already deleted “with the results detailed in”.

Comments 27: Line 100: The statement “These two toxin proteins were absent in the non-Bt maize tissues” conflicts with “There were significant differences in the expression levels of Cry1Ab, Vip3Aa, and total Bt protein across the tissues.” Please clarify.

Response: As suggested, we have clarified the statements in the manuscript. The revised sentences are now described as: “ELISA results confirmed that non-Bt maize tissue lacked the Cry1Ab or Vip3Aa insecticidal proteins. In DBN3601T Bt maize tissues at various growth stages, the concentrations of Cry1Ab, Vip3Aa, and total Bt protein ranged as follows: 1.71~78.93 μg/g, 0.68~29.27 μg/g, and 2.39~108.20 μg/g, as presented in Table 1. Significant variations were observed in the expression levels of Cry1Ab, Vip3Aa, and total Bt protein across different tissues of DBN3601T maize”. (lines 109-114)

Comments 28: Line 112: Change “Susceptibilities” to “Susceptibility.”

Response: As suggested, we have changed “Susceptibilities” to “Susceptibility”. (line 124)

Comments 29: Line 127: What do the authors mean by “corrected” mortality rates? Clarify here and in other places.

Response: Revised. We clarified this in the “Materials and Methods” section as follows “Mortality rates were calculated after 7 days. To ensure an accurate assessment of the insecticidal effects, mortality rates were corrected using Abbott's formula to account for natural mortality observed in the control group [62].” (lines 411-414)

Comments30: Lines 129-130: Refer to the abstract comments about 100.00%.

Response: Revised.

Comments 31: Line 154: Use consistent and concise terminology. Writing “Bt-(Cry1Ab+Vip3Aa) maize” every time is cumbersome. Provide detailed information in the methodology section and later use Bt-maize” and “non-Bt maize.”

Response: Revised. We replaced "Bt-(Cry1Ab+Vip3Aa) maize" with "DBN3601T maize" throughout the manuscript.

Comments 32: Lines 159-160: If discussing control, talk about the reduction in plant damage, not the increase.

Response: Revised. The sentence has been revised to “Control efficacy, measured by the reduction in plant damage rate, was 82% on the first day and reached 100% on the eleventh day.” (lines 169-171)

Comments 33: Line 169: Delete “(D)” from “Different lowercase letters (D) indicate.”

Response: Revised. We have removed “D”.

Comments 34: Discussion: This section seems to repeat the results and does not provide a summary of key findings or interpret the findings in the context of existing literature.

Response: Accepted. We have thoroughly revised the Discussion section, removing redundancies and summarizing key findings. Please refer to the updated Discussion section.

The modifications are as follows: “Our study indicates that Cry1Ab and Vip3Aa proteins are particularly effective against S. exigua, with Cry1Ab showing high biological activity. Previous studies have shown that Cry1Da and Cry1Ca proteins demonstrated strong lethal and growth-inhibitory effects on S. exigua, while Cry1Ba and Cry2Ab proteins are non-toxic against this pest [37]. In the field, Bt-Cry1F cotton can effectively control the damage of S. exigua, while Bt-Cry1Ac cotton requires the application of insecticides to mitigate its damage, mainly because S. exigua is more susceptible to Cry1F protein [38,39]. Compared to lepidopteran pests such as O. furnacalis and S. frugiperda, Cry1Ab and Vip3Aa proteins exhibit stronger lethality against S. exigua in Henan and Anhui Provinces [34,35,40], further confirmed that Bt maize expressing Cry1Ab or Vip3Aa proteins holds the potential for controlling S. exigua.” (lines 186-196)

Comments 35: Line 174: The names Lepidoptera, Coleoptera, Diptera are orders. If referring to pests, use Lepidopteran,” “Coleopteran,” “Dipteran” or “pests belonging to the orders Lepidoptera, Coleoptera, Diptera.” Update the text accordingly.

Response: As suggested, the phrase is now described as “such as Lepidopteran, Coleopteran, Dipteran, etc.” (line 184)

Comments 36: Line 179: “Curb its damage” curb is not a suitable word here.

Response: Revised. We have replaced “curb” with “mitigate”. (line 192)

Comments 37: Lines 183-184: The discussion section is not ideal for mentioning data values.

Response: As suggested, we have revised the discussion section to remove specific data values.

Comments 38: Lines 185-186: Repeating the location of S. exigua larvae from Pu’er City, Yunnan Province, is unnecessary; it was already mentioned in the methodology.

Response: As suggested, we have deleted the redundant part.

Comments 39: Lines 194-195: This part is unclear.

Response: Revised. The sentence is now described as “The pest-controlling effect of Bt maize mainly depends on the expression of its insecticidal proteins across its different tissues and growth stages [26]” (lines 200-201)

Comments 40: Line 195: Use “findings” or “observations” instead of “measurements.”

Response: As suggested, we have changed “measurements” to “observations”. (line 201)

Comments 41: Line 227: Numbers are not needed here; there is a dedicated section for this in the results.

Response: As suggested, the numerical data has been removed.

Comments 42: Conclusion: The conclusion should be improved to provide a take-home message, emphasizing what this study contributes to the field that was not known before.

Response: Accepted. As suggested, the revised conclusion is as follows: “This study demonstrated that Bt-(Cry1Ab+Vip3Aa) maize DBN3601T can efficiently control the lepidopteran pest S. exigua, providing a new measure for the sustainable management of this pest in Asia. Planting of DBN3601T, even on a subset of cropping areas, may facilitate area-wide management of S. exigua and other polyphagous lepidopteran pests, as has been observed in cotton systems [64,65]. This can be achieved by integrating existing IPM strategies, such as pest monitoring and early warning, agriculture practices (e.g., reasonable rotation and intercropping), physical control (e.g., light traps), and biological control (e.g., parasitoid releases, sex pheromone traps, applications of baculoviruses and conservation biological control schemes [6,66]. Consequently, pest populations can be suppressed over a wide geographic area, leading to sustainable intensification of maize crops.” (lines 443-453)

Round 2

Reviewer 2 Report

Comments and Suggestions for Authors

The authors have adequately responded to this reviewer's comments and suggestions

Reviewer 4 Report

Comments and Suggestions for Authors  

The current version has substantially improved, and the authors have incorporated my suggestions. The manuscript can be accepted.